# Estimation of Evaporation of Water from a Liquid Desiccant Solar Collector and Regenerator by Using Conservation of Mass and Energy Principles

Gezahegn Habtamu Tafesse [1,*], Gulam Mohammed Sayeed Ahmed [1,2], Irfan Anjum Badruddin [3], Sarfaraz Kamangar [3] and Mohamed Hussien [4]

1   Department of Mechanical Engineering, School of Mechanical, Chemical and Materials Engineering, Adama Science and Technology University, Adama P.O. Box 1888, Ethiopia
2   Center of Excellence (COE) for Advanced Manufacturing Engineering, Program of Mechanical Engineering, School of Mechanical, Chemical and Materials Engineering, Adama Science and Technology University, Adama P.O. Box 1888, Ethiopia
3   Mechanical Engineering Department, College of Engineering, King Khalid University, Abha 61421, Saudi Arabia
4   Department of Chemistry, Faculty of Science, King Khalid University, Abha 61413, Saudi Arabia
*   Correspondence: gezahegn.habtamu@astu.edu.et or gezisha@yahoo.com

**Abstract:** Solar thermal energy-powered air conditioning technologies are receiving increased attention. Among the solar energy-driven cooling technologies, open type liquid desiccant air conditioning (AC) system is emerging as a promising technology, which has a solar driven desiccant solution regenerator. In this type of system, the evaporation of water and concentrating the desiccant or regenerator performance determines the cooling performance of the AC system, which necessitates its development and experimental performance testing under actual operating conditions. The setup is made of a black painted corrugated solar collector of area 0.8 m × 1.84 m covered with glass, and a liquid desiccant solution tank and distribution system over the absorber. Solar regeneration experiments on calcium chloride–water solution were carried out on the setup and a total of five sets of meteorological, collector and solution property data were collected through concentrating the desiccant from 32.9 initially to 51.3% in five days. The evaporation of water from the regenerator was analyzed using energy and desiccant mass conservation. For a typical day, the mass of water evaporated was estimated to be 3.10 and 3.16 kg over a day, as estimated by conservation of mass and energy principles from a 34.8 kg of calcium chloride solution with initial desiccant concentration of 43.6% stored in the tank.

**Keywords:** solar thermal energy; evaporation of water; regenerator; liquid desiccant; calcium chloride; conservation of mass and energy

## 1. Introduction

Air conditioning systems are primarily needed to supply conditioned or processed outdoor air to buildings after reducing the temperature and moisture to the desired humidity ratio and temperature. This is achieved commonly by conventional vapor compression air conditioning systems by the process of simultaneously cooling air below the dew point temperature of moisture in the air. Vapor compression systems are effective but use environmentally unfriendly refrigerants and are not energy efficient since they consume large amount of high-grade electricity. Desiccant based air conditioning systems cool air without the need of cooling the air below the dew point temperature of air [1], and they are an alterative system that uses an environmentally friendly refrigerant with greater potential to be powered by low-grade energy such as solar or waste heat with minimal electricity use. There are two major categories of this system: solid desiccant and liquid desiccant systems. These systems use desiccant materials which have different affinities of absorbing

and desorbing moisture. In both systems, air condition is achieved by dehumidifying outdoor air using strong desiccant material and regenerating the desiccant material by evaporating the absorbed moisture, which can be regenerated using different methods such as electrodialysis, thermal air and mechanical vapor compression [2], indirect solar regeneration [3] and direct solar regeneration [4]. In a solid desiccant system, a dry rotary desiccant disc removes moisture from outdoor air and the wet disc is then dried or regenerated using heat for reuse. It is also efficient in controlling supply air temperature and humidity and can be driven by thermal energy collected by flat plate, evacuated tube and compound parabolic solar collectors for regenerating the disc [5]. Silica gel, activated alumina, lithium chloride, natural zeolites and molecular sieves desiccants are commonly used in solid desiccant systems [5,6]. Liquid desiccant-based air conditioning technology is an emerging technology for use in hot and humid climatic conditions for summer air conditioning and also winter air space heating [7]. It uses a desiccant solution as a refrigerant to satisfy the two AC requirements: dehumidification and sensible cooling of outdoor air. The common liquid desiccant materials are calcium chloride, lithium chloride and lithium bromide [1]. Hybrid desiccant based air conditioning systems technologies are also emerging. Kumar et al. [8] studied lithium chloride and potassium formate-based liquid desiccant and vapor compression hybrid systems and Naik et al. [9] studied the dehumidification performance of a solid desiccant made of silica gel and a lithium chloride liquid desiccant hybrid air dehumidification system for dehumidifying fresh air and for drying applications. Heat for regenerating the liquid desiccant can be obtained from waste heat [10] and solar energy [3]. There are two types of solar-driven liquid desiccant air conditioning system, depending on the desiccant regeneration technology: indirect regenerator, which uses water as solar energy collecting fluid in evacuated tube solar collectors [3] for heating the liquid desiccant in a heat exchanger; and direct regenerator, which uses the liquid desiccant as solar energy collecting fluid in an open-type flat plate collector [4]. The direct regeneration-type solar liquid desiccant AC system has two essential system units that must work cyclically to perform dehumidification of the fresh air: *dehumidifier*, where dehumidification of fresh air is conducted by strong desiccant solution; and *solar regenerator*, where weak desiccant solution (strong desiccant solution + air moisture) is heated by solar energy to reproduce or regenerate the strong desiccant solution. The regenerator collects solar thermal energy for heating the desiccant solution and desorbs the water from the weak solution in the solar collector itself, which avoids the need of a desorption tower and a blower [1]. Thus, it is both a heat and mass transfer device; hence, the name "solar regenerator". For every kilogram of water desorbed in the solar regenerator, 1 kg of water can be dehumidified in the air dehumidifier. This makes the solar regenerator a vital component of open type liquid desiccant cooling technology since the evaporation performance of the solar regenerator determines the cooling performance of the cycle [11]. Several studies were conducted on the regeneration of weak desiccant solution using solar energy for estimating the evaporation performance. Mullick and Gupta [4] performed detailed experimental and theoretical analysis on solar regeneration of calcium chloride solution. The evaporation efficiency of the collector was of the order of 40% for irradiation of 0.93 kW/m$^2$, 50% CaCl$_2$ solution concentration, 38 °C outdoor air temperature, and collector temperature of 65 °C. Collier [11] developed an analytical procedure for calculating the mass of water evaporated from weak desiccant solution in an open flat plate collector. Gandhidasan [12] developed a simplified equation to estimate the rate of evaporation of water from the weak solution for a closed-type solar still regenerator. Peng and Howell [13] compared numerical and analytical evaporation models for open inclined surface solar regenerators. Haim et al. [14] performed simulation and analysis of direct and indirect regeneration or evaporation of water from lithium chloride solution. The study presented evaporation performance curves of the regenerator under different operating parameters and concluded that the direct regeneration system is better than the indirect one. Yang and Wang [15–18] conducted an experimental study on the regeneration of the desiccant solution and reported that counter flow of scavenging air improved the efficiency of the system; a forced convection ventilated

double-glazed solar regenerator evaporation performance was found to be better than the single-glazed equivalent. In addition, their study revealed an optimum value of 7 cm glazing gap. Hawlader et al. [19] conducted experimentation on an 11 m × 11 m unglazed collector and regenerator. The experimental results show a regeneration efficiency varying between 38% and 67%. Alizadeh and Saman [20,21] developed a computer model to evaluate the thermal performance of the solar collector and regenerator. They used a forced parallel flow prototype of a solar collector and regenerator and conducted experimentation. The test results revealed that flow rates of ventilation air and solution mass and the meteorological conditions affect the regeneration. Kabeel [22] carried out free and forced convection studies on two separate solar collectors/regenerators and found that the forced convection resulted in better regeneration efficiency. However, Elsarrag [23] experimentally studied the effects of ratio of solution to air flow rates, the humidity ratio of air and the temperature and concentration of the desiccant in the solution on the evaporation rate of calcium chloride solution in a corrugated plate collector of 1 m × 1 m. The optimum value of the liquid to air flow rate was reported as ≤2.54. Yutong and Yang [24] identified the effects of air and solution conditions on the heat and mass transfer coefficients by conducting indoor regeneration of lithium bromide (LiBr) brine in a laboratory using a solar simulator. Previous experimental research efforts on evaluating the concentrating performance of the solar regenerator uses two solution tanks for separately measuring the inlet concentration of the desiccant coming from the air dehumidifier and outlet concentration of the desiccant leaving the regenerator for the dehumidifier and applies the conservation of desiccant mass and energy principles over the absorber surface to estimate the evaporation in the regenerator. Energy is conserved but mass is not conserved over the absorber area in this two-tank experimental procedure. In a two-tank experimental regeneration procedure, the desiccant mass over the absorber surface varies along the length and width of the collector as it flows from inlet to the outlet. Moreover, the resident time of the flowing solution over the absorber is very small and it is difficult to detect measurable change in concentration at exit from the collector. Both the mass flow rate of the solution and the desiccant over the absorber surface fluctuate and measuring the concentration at a point towards the exit will not represent the final concentration of the solution. The better experimentation procedure is the use of a single-solution tank as control volume for applying conservation of desiccant mass in the tank over an appreciable change in time and applying conservation of energy over the absorber to evaluate how the solar thermal energy is utilized. In addition, use of a single-solution tank helps in decoupling the solar regenerator from the air dehumidifier cycle and provides flexibility in regenerating the weak solution anywhere around a building where solar energy is available. More evaporation happens as a result of more solar and thermal energy absorption when the desiccant material stays on the solar absorber through recirculation. The solar to thermal energy conversion ability of the regenerator depends on the optical and thermal characteristics of its constituent parts and the concentration of the desiccant. Thus, the combination of conservation of mass and energy principles helps in the design of the optimized solar regenerator area in relation to the desiccant solution volume and initial concentration. This paper focuses on estimating the experimental evaporation performance during the outdoor regeneration of water and calcium chloride solutions in a glazed solar regenerator by direct application of conservation of desiccant mass and energy principles using desiccant solutions of known initial concentration stored in single tank.

## 2. Materials and Methods

### 2.1. Experimental Setup and Instruments

The solar regenerator system consists of a glazed, corrugated, galvanized sheet, painted black, with length 0.8 m and width 1.84 m, a concentric pipe solution distributor of effective length 1.75 m (the outer pipe is slotted along its length and the inner pipe contains a series of small holes every 20 mm distance along the length), a solution gutter at the end of the sheet, a supply tank and a centrifugal pump, as shown in Figure 1. The regenerator bottom and edges are insulated with glass wool of 102 mm and 20 mm

thickness, respectively. The glass cover used was 3 mm window glass. The collector was inclined at 14° and the glazing gap was 8.5 cm. The size of the solution tank was 57 cm × 57 cm × 33 cm, which was made of polyethylene materials and a small pump was used to circulate the weak solution over the absorber surface through the distributor. The absorber was painted iron oxide and then black paint to control corrosion and absorb the solar energy. The instruments used to collect the experimental data include the following: T-type thermocouples (installed on the surface of the absorber in 4 rows), 4 RTD (used to detect outlet and inlet dry and wet bulb temperatures of the scavenging air), hand thermometer, anemometer, portable densitometer and pyranometer. The readings of the thermocouples, the RTDs and the pyranometer were collected every minute via a data acquisition system (data logger) interfaced with a PC under the weather conditions of Delhi. The sensitivity of these instruments is indicated in Table 1.

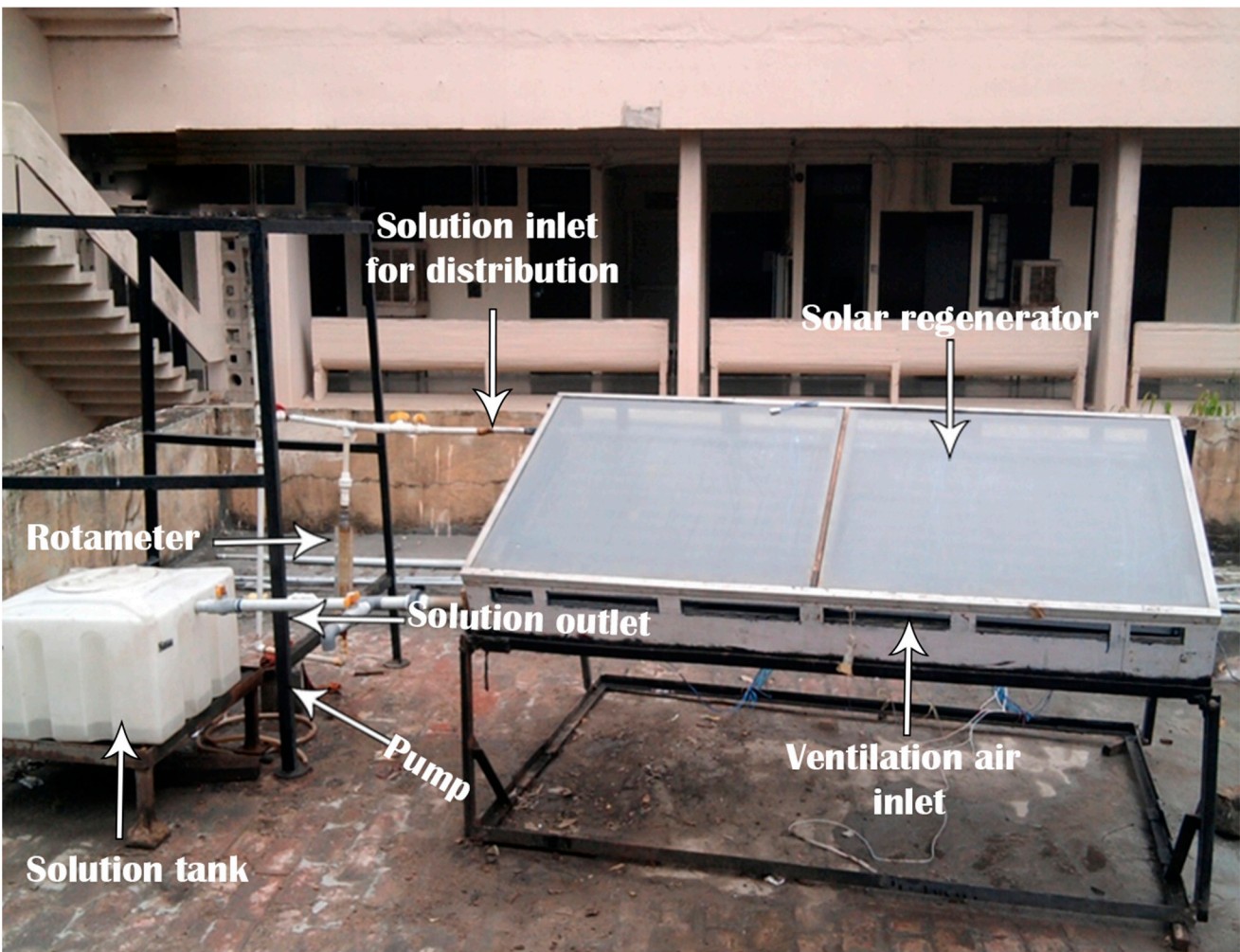

**Figure 1.** The Solar thermal liquid desiccant regenerator experimental setup.

**Table 1.** Uncertainty of measuring instruments.

| S. No. | Instrument | Uncertainty |
|---|---|---|
| 1 | Densimeter | $\pm0.001$ g/cm$^3$ |
| 2 | Pyranometer | $\pm3\%$ of pyranometer reading |
| 3 | Thermometer | $\pm0.5$ °C |
| 4 | Thermocouple wire | $\pm0.5$ °C |
| 5 | RTD | $\pm0.5$ °C |
| 6 | Tape rule/Scale | $\pm1$ mm |

### 2.2. Working Principle of the Solar Regenerator

Regeneration experimentation was carried out first by storing calcium chloride desiccant and water solution in the tank. The data logger that collects the surface temperature of the absorber, glass and solar irradiance will be switched on. Then, the initial solution temperature and density was recorded using a thermometer and hand-held densitometer (Figure 1), respectively. These measurements were used to determine the initial mass fraction of the desiccant in the solution which will remain constant since it is nonvolatile during regeneration. Since the experimentation usually starts a few minutes before 9 am, the temperature of the absorber surface is much higher than the temperature of the solution in the tank due to the absorbed solar radiation. After determining the initial mass of the solution and desiccant concentration in the tank, the solution will be circulated over the absorber surface to bring the thermal inertia of the solution in the tank and the absorber surface to thermal equilibrium. The solution that comes from the solar regenerator is mixed with the solution that remains in the tank, as seen in Figure 2. After this, the new solution density and the temperature are recorded; at this time, the solution temperature increases due to two reasons: heat gain from the absorber and because the desiccant dehumidifies the atmosphere air. Then, measurements of solution density and temperature in the tank are repeated every thirty minutes till the end of experimentation to monitor the thermodynamic state change of the solution using the tank as control volume.

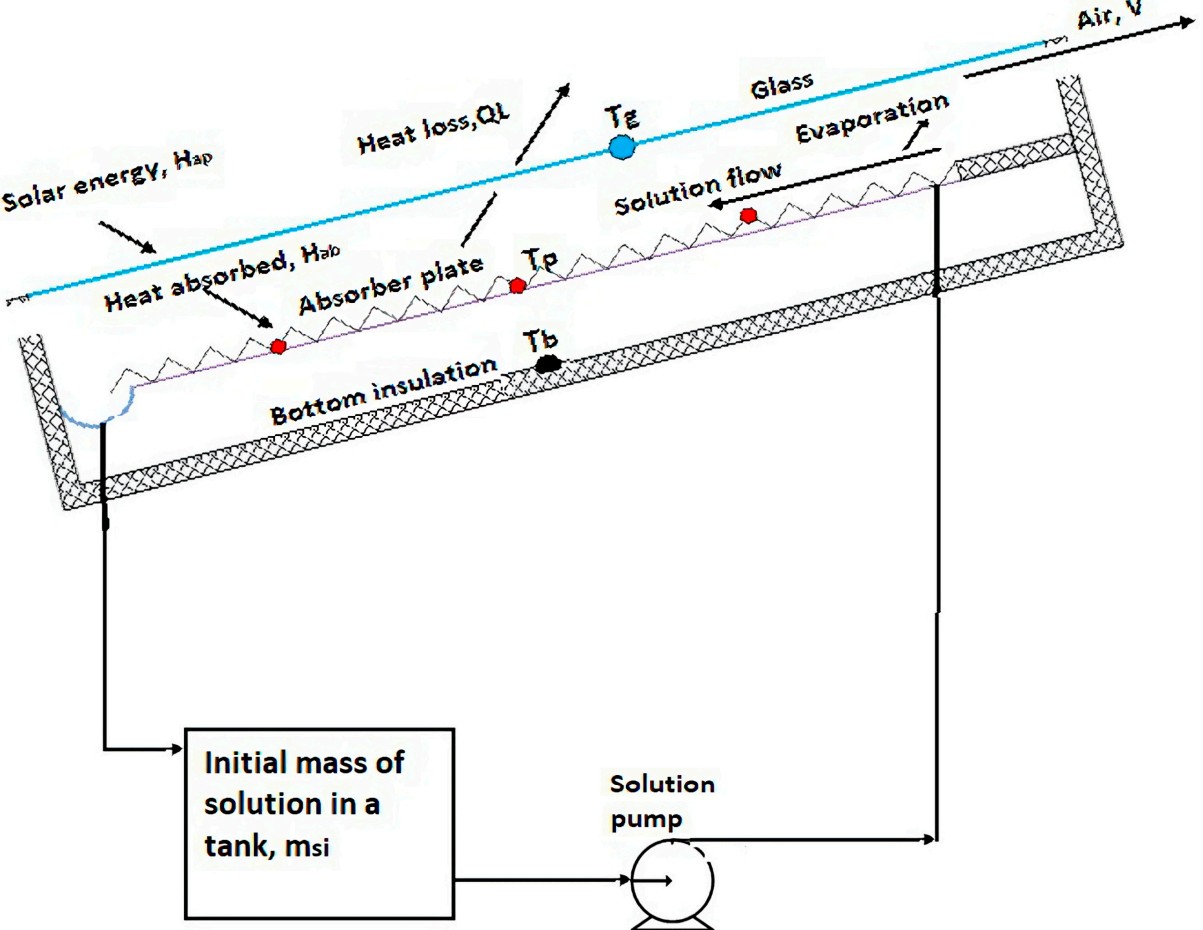

**Figure 2.** The schematic diagram of the solar regenerator working principles.

### 3. Data Reduction

Solar heating of a weak liquid desiccant solution in the regenerator causes a rise in temperature, thereby increasing the vapor pressure of water in brine. When the pressure of the water in the solution is greater than the partial pressure of water in the ambient air,

energy and mass transfer take place from the weak solution due to water evaporation in the solar regenerator [11,12]. The mass of water evaporated can be estimated mainly using conservation of mass, and approximately using conservation of energy as discussed below.

### 3.1. Estimation of Water Evaporation Using Conservation of Desiccant Mass

The concentration of the calcium chloride desiccant in the solution (in the range of 0 to 0.6) can be determined using measured values of solution density and solution temperature from the following equation [25]:

$$0.105642\left(\frac{\xi}{1-\xi}\right)^3 - 0.4363\left(\frac{\xi}{1-\xi}\right)^2 + 0.836014\left(\frac{\xi}{1-\xi}\right) + 1 - \frac{\rho_{sol(\xi,T)}}{\rho_{H_2O(T)}} = 0. \quad (1)$$

The density of water at the solution temperature can be determined from [26]:

$$\rho_{H_2O(T)} = 1000\left(1 - \left(\frac{T + 288.9414}{508929.2(T + 68.12963)}\right)(T - 3.9863)^2\right). \quad (2)$$

The evaporated water mass from the solution, based on conservation of mass, is the difference between the initial and the final mass of solution in the tank [27]:

$$m_v = m_{si} - m_{sf}. \quad (3)$$

Since desiccant is a non-volatile material; the initial and final mass of the desiccant in the solution remains constant. In terms of solution mass, it is expressed as:

$$m_{si}\xi_i = m_{sf}\xi_f \Rightarrow m_{si}\xi_i = (m_{si} - m_v)\xi_f. \quad (4)$$

The mass of vapor evaporated for any time interval can be obtained by replacing, initial concentration with $\xi_i = \xi_{t_o+(n-1)\Delta t}$, final concentration with $\xi_f = \xi_{t_o+n.\Delta t}$ and initial mass of the desiccant in the tank before starting experiment $m_{si}\xi_i = m_d$. Rearranging Equation (4) yields the evaporated water mass as [27]:

$$m_v = m_{si}\xi_i\left(\frac{1}{\xi_i} - \frac{1}{\xi_f}\right) = m_d\left(\frac{1}{\xi_{t_o+(n-1).\Delta t}} - \frac{1}{\xi_{t_o+n.\Delta t}}\right) \text{ where } n = 1, 2, \ldots. \quad (5)$$

Solar Regenerator Efficiency

Solar collection cum regeneration efficiency is defined as [22]:

$$\eta_R = \frac{E_v}{H_{ap}}. \quad (6)$$

The fraction of solar energy consumed to evaporate water from the desiccant solution or total energy used for evaporation can be estimated from:

$$E_v = m_v h_{fg}. \quad (7)$$

For calcium chloride and water mixture, the heat of vaporization for water from the solution is taken as the addition of latent heat of pure water vaporization and the enthalpy of dilution of calcium chloride desiccant [25].

$$h_{fg} = 2551 + \Delta h_d$$
$$= 2551 + (-955.69 + 3011.974\theta)\left[1 + \left(\frac{\xi}{0.855(0.8-\xi)}\right)^{-1.965}\right]^{-2.265} \quad (8)$$

The solar energy striking the regenerator aperture area is [28]:

$$H_{ap} = A_{ap}I.\Delta t \ (kJ). \quad (9)$$

### 3.2. Estimation of Water Evaporation Using Conservation of Energy

Taking the solar absorber and the thin liquid desiccant solution flowing over it as the thermodynamic system boundary, the energy influx and outflow into and out of the regenerator system are absorbed solar radiation, energy losses by heat, the energy convected out by the scavenging air passing between the regenerator glass and the absorber solution, the energy stored in the desiccant solution and the energy transferred out of the regenerator as a result of the mass of water evaporated. The thermal resistance network diagram for the solar regenerator is indicated in Figure 3.

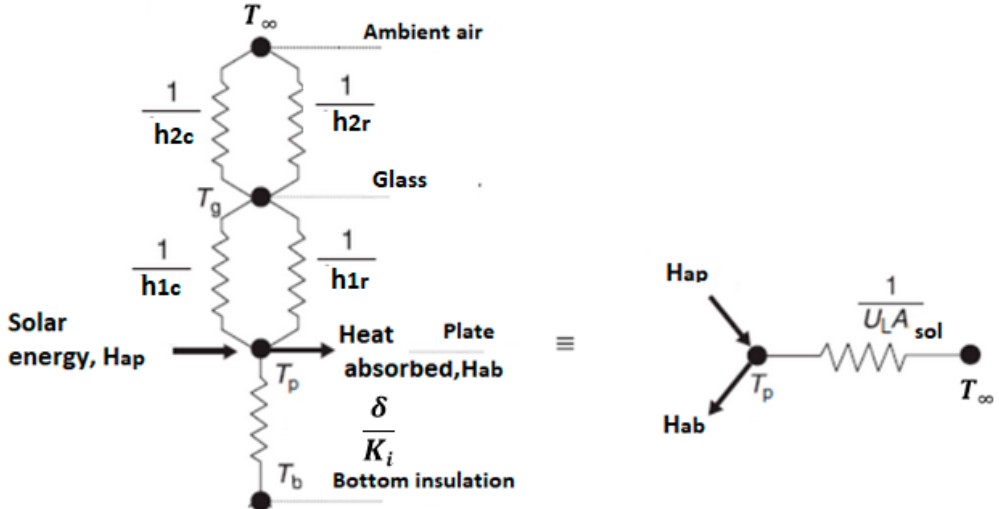

**Figure 3.** The thermal resistance network diagram for the solar regenerator.

Assuming steady state condition, application of the conservation of energy principle to the solar regenerator yields:

$$H_{ab} - U_L A_{sol}(T_P - T_\infty).\Delta t - VA_{srai}\rho_a C_{pa}(T_{ao} - T_{ai}).\Delta t - \\ \left\{ (\dot{m}_{si} - \dot{m}_v)C_{pso}T_{so} - \dot{m}_{si}C_{psi}T_{si} \right\}.\Delta t - m_v h_{fg} = 0 \tag{10}$$

Rearranging Equation (10) yields an approximate equation that can be used to determine the mass of water evaporated as:

$$m_v = \frac{H_{ab} - U_L A_{sol}(T_P - T_\infty).\Delta t - VA_{sraia}\rho_a C_{pa}(T_{ao} - T_{ai}).\Delta t - m_{si}(C_{Pso}T_{so} - C_{Psi}T_{si})}{h_{fg} - C_{Pso}T_{so}}. \tag{11}$$

The absorbed solar incident energy by the collector area is given by:

$$H_{ab} = \eta_{opt}I.A_{sol}.\Delta t. \tag{12}$$

The optical efficiency of the solar regenerator is give by:

$$\eta_{opt} = (1-s)(1-d)\tau\alpha. \tag{13}$$

The total heat loss coefficient $(U_L)$ is the sum of the heat loss from the top $(U_t)$, bottom $(U_b)$ and sides $(U_s)$ of the collector [29]:

$$U_L = U_t + U_b + U_s. \tag{14}$$

The top loss coefficient $(U_t)$ can be estimated by considering convection heat losses (from collector to glass cover $(h_{1c})$ and glass cover to ambient $(h_{2c})$) and radiation heat

losses (from the collector to glass cover ($h_{1r}$) and from glass cover to ambient ($h_{2r}$)) in the upward direction. Thus, the effective top heat loss coefficient is given by [29]:

$$U_t = \left[ \frac{1}{h_{1c} + h_{1r}} + \frac{1}{h_{2c} + h_{2r}} \right]^{-1};$$

(15)

$$h_{1c} = \frac{K_a}{L} Nu.$$

(16)

The Nusselt number in terms of the Rayleigh number for collector inclined between 0 to 75° as [29]:

$$Nu = 1 + 1.44 \left[ 1 - \frac{1708 \sin(1.8\beta)^{1.6}}{R_a \cos \beta} \right] \left[ 1 - \frac{1708}{R_a \cos \beta} \right]^{+} + \left[ \left( \frac{R_a \cos \beta}{5830} \right)^{1/3} - 1 \right]^{+}.$$

(17)

"+" exponent means only the positive value of the square bracket is to be used. Zero is to be used for negative value. The Rayleigh number is given by:

$$R_a = \frac{2g(T_p - T_g)\text{Pr}L^3}{(T_P + T_g)v^2};$$

(18)

$$h_{2c} = 5.7 + 3.8V;$$

(19)

$$h_{1r} = \sigma \left[ \frac{1}{\varepsilon_p} + \frac{1}{\varepsilon_g} - 1 \right]^{-1} \left[ \frac{(T_p + 273)^4 - (T_g + 273)^4}{T_p - T_g} \right];$$

(20)

$$h_{2r} = \varepsilon_g \sigma \left[ \frac{(T_g + 273)^4 - (T_\infty + 267)^4}{T_g - T_\infty} \right].$$

(21)

Neglecting the bottom convective heat loss and assuming one dimensional steady state heat flow through the bottom insulation, the bottom heat loss coefficient ($U_b$) is expressed as:

$$U_b = \frac{K_i}{\delta_b}.$$

(22)

The side loss coefficient ($U_s$) is given by:

$$U_s = \frac{Ld}{A_{sol}} \frac{K_i}{\delta_s}.$$

(23)

The specific heat capacity at constant pressure for calcium chloride water solution is a function of concentration and water specific heat at the solution temperature. It is given by [25]:

$$C_{Psol(\xi,T)} = C_{PH_2O(T)}[1 - f_1(\xi)f_2(T)];$$

(24)

$$C_{PH_2O(T)} = 88.789 - 1201958\theta^{0.02} - 16.9264\theta^{0.04} + 52.4654\theta^{0.06} + 0.10826\theta^{1.8} + 0.46988\theta^8;$$

(25)

$$f_1(\xi) = 1.63799\xi - 1.69002\xi^2 + 1.05124\xi^3;$$

(26)

$$f_2(T) = 58.5225\left(\frac{T}{228} - 1\right)^{0.02} - 105.6343\left(\frac{T}{228} - 1\right)^{0.04} + 47.7948\left(\frac{T}{228} - 1\right)^{0.06};$$

(27)

$$\theta = \frac{T}{228} - 1.$$

(28)

As per the manufacturer's guideline, the densitometer reading was calibrated with pure water density. General uncertainty analysis was carried out for estimating the uncertainty of calculated performance indicators of the regenerator using [30]:

$$U_R^2 = \sum_i^N \left( \partial R \Big/ \partial x, i * U_{x,i} \right)^2. \tag{29}$$

The uncertainties of pyranometer, densitometer and RTD/thermocouple wires were 3% of pyranometer reading, 0.001 g/cm$^3$ and 0.5 °C, respectively. The uncertainty of tape rule/scale was 1 mm. The estimated solution volume uncertainty from the scale measurement was 0.625 L. The maximum uncertainty over a one hour period in calculating the increase in concentration, the water evaporated based on the conservation of mass principle and solar regenerator efficiency were obtained as 0.0014 kg/kg, 0.05 kg and 0.02, respectively.

## 4. Results and Discussion

Among the five experiments conducted on the solar regenerator in the summer months of late March, one complete 1 h average data collected for a typical day in March (day 3, the one with recommended calcium chloride concentration for air conditioning application) is presented to illustrate the estimation of the evaporation performance of the regenerator by applying the mass and energy conservation equation(s) as discussed above. Some of the essential data include 34.8 kg initial mass of calcium chloride–water solution ($m_{si}$) in the tank; initial concentration 43.6 kg/kg; collector inclination 14°; volume flow rate 10 lpm; shade coefficient $s = 0.02$; dust coefficient $d = 0$; Collector absorptivity $\alpha = 0.9$; glass transmissivity $\tau = 0.85$; regenerator aperture area $A_{ap} = 1.92$ m × 0.9 m; solar area $A_{sol} = 1.84$ m × 0.8 m; solar regenerator air inlet area (sraia) = 1.84 m × 0.04 m; wind speed V = 0.4 m/s; and calculated overall heat transfer coefficient of the collector $U_L = 13.7$ W/m$^2$K. These data are shown in Table 2.

**Table 2.** Characteristics of the solar regenerator system and initial data.

| S. No. | Parameter | Values |
|---|---|---|
| 1 | Initial solution mass ($m_{si}$) | 34.8 kg |
| 2 | Initial concentration ($\xi_i$) | 43.6 kg/kg |
| 3 | Solution tank size | 57 cm × 57 cm × 33 cm |
| 4 | Collector areas (apearure and absorber) | 1.92 m × 0.9 m and 1.84 m × 0.82 m |
| 5 | Collector inclination | 14° |
| 6 | Wind speed, | 0.4 m/s |
| 7 | Glass transmisivity | 0.85 |
| 8 | Absorber plate absoptance | 0.9 |
| 9 | Shade coefcient | 0.02 |
| 10 | Collector heat loss coefcient | 13.7 W/m$^2$ K |
| 11 | Volume flow rate of solution | 10 lpm |

Figure 4 (top left) shows the variation of temperatures of solution, mean absorber plate, regenerator inlet and outlet air temperatures, solution density, irradiance measured and the calculated desiccant concentration throughout the day. The result shows that the temperature of the solution is slightly greater than the absorber temperature from 10:00 am to 11:00 am, after which the temperature of the absorber becomes slightly greater than the solution temperature. The average maximum temperature discrepancy between the solution and the collector is 2 °C; as a result, the solution and the collector temperature can be considered the same. The glass surface temperature is almost the same as the outlet

temperature of the scavenging air from the regenerator. The mean absorber plate temperature throughout the day was obtained after measuring the absorber surface temperature by using nine T-type thermocouples fixed on to three rows across the width and 30 cm distance above and below the center of the absorber length. The minimum and maximum collector temperature variations detected by the thermocouples were 2 and 5 °C, respectively.

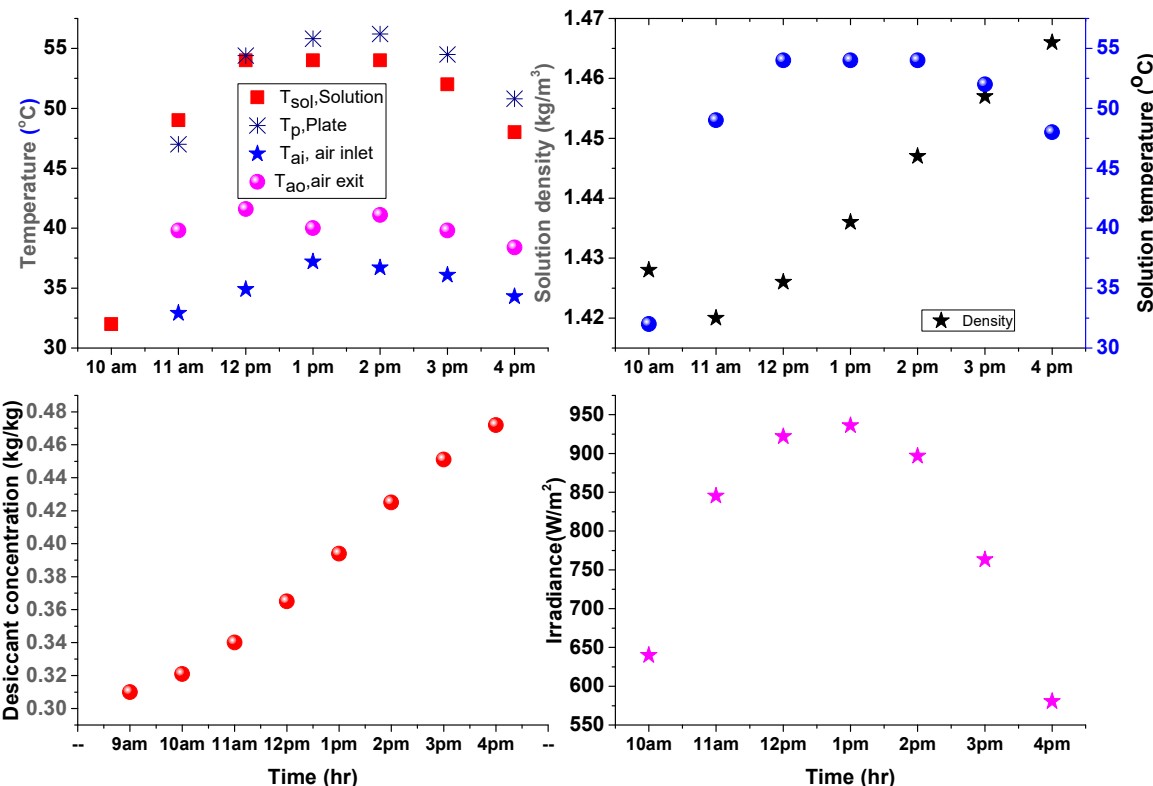

**Figure 4.** Experimentally measured temperatures, irradiance, solution density and calculated concentration with time.

The measured density and temperature of the solution are depicted in Figure 4 (top right) and the density slightly decreases for about an hour and start to increase afterwards whereas the solution temperature varies from 10 am to 4 pm with peak value of 52 °C around noon. The desiccant concentration was calculated by Equation (1) after measuring the solution temperature and density. As seen in Figure 4 (bottom left), the desiccant concentration in the solution increased linearly with time from 43.6% to 47.9%, agreeing well with Hawlader [19]. The total rise in concentration of the desiccant in the solution within the day was 9.7%. The solar flux received by the regenerator (bottom right of Figure 4) was high as the day was clear sky with day average value of about 807.1 W/m². The trend of solution temperature and the solar irradiance are similar. In Figure 5, the trend of evaporation of water mass from the desiccant solution as estimated by conservation of mass, or Equation (5), and conservation of energy, or Equation (11), is more or less same in trend and magnitude for this test, and in both cases, it is peaking between 12:30 pm and 1:30 pm. The total amount of water evaporated from the solution within the day using Equations (5) and (11) is 3.10 kg and 3.16 kg, respectively. This estimation shows that the prediction of mass of water evaporated by the conservation of mass was more or less the same as the prediction of the conservation of energy equation, which involves several empirical relations on the heat transfer, specific enthalpy of evaporation and optical characteristics of the solar regenerator that should be improved for better estimation. Figure 5 also shows the variations of the regeneration efficiency curves which follow the pattern of the variation of the irradiance striking the absorber or energy utilized in evaporating water from the solution by the regenerator. The result indicates that solar

collection and regeneration or regenerator efficiency varies from 6.0 to 37.1% when using conservation of mass (or Equation (5)).

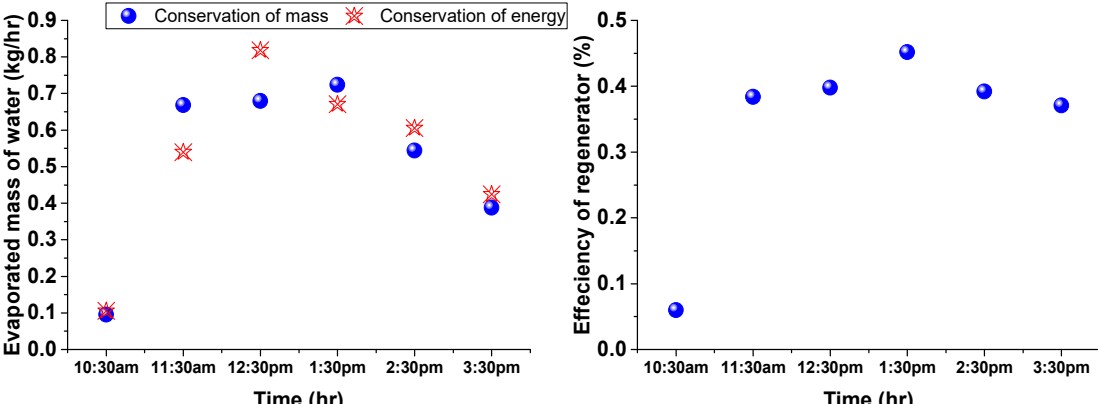

**Figure 5.** Variation of mass of water evaporated as calculated by conservation of mass and energy principles with time.

The intermediate data needed for the estimation of the evaporation rate of water using conservation of energy equation (Equation (10)) are indicated in Figures 6 and 7.

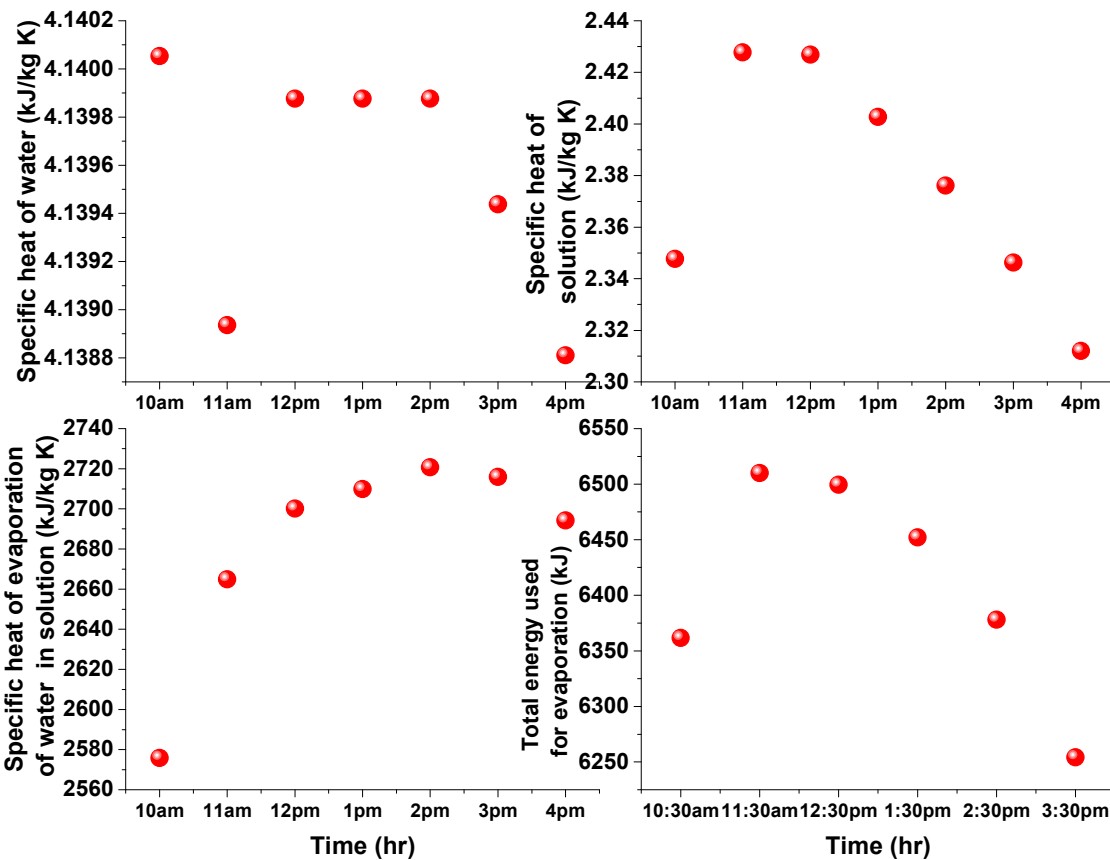

**Figure 6.** Variation of the specific heat of water the specific heat of solution, the specific heat of evaporation of water in the solution and the total energy of evaporation of water from the solution with time.

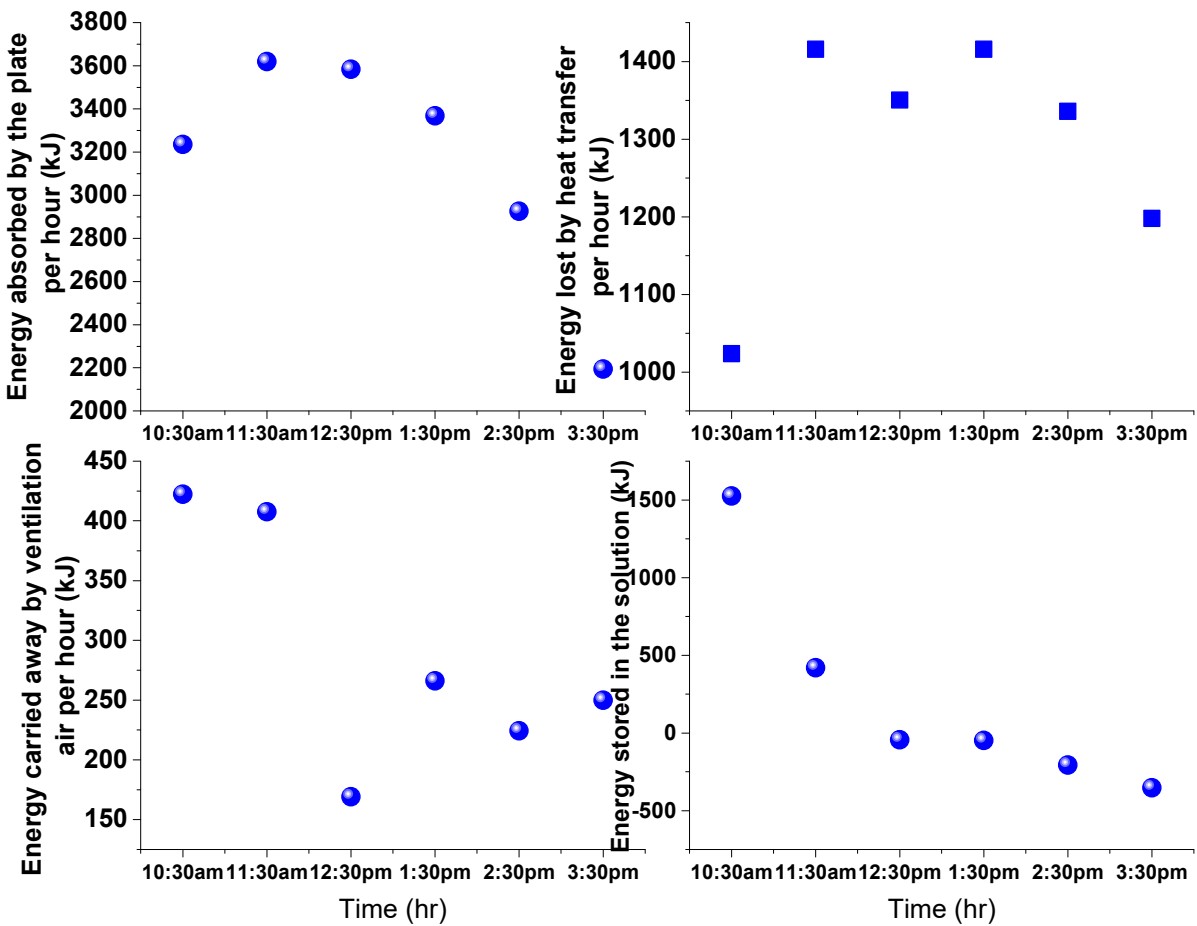

**Figure 7.** Variation of energy parameters needed to estimate the evaporation rate of water using the conservation of energy principle over time.

Figure 6 shows the specific heat of pure water and desiccant solution, the latent heat of evaporation of water from the desiccant solution and the heat transferred as a result of evaporation from the regenerator. Figure 7 demonstrates the total solar energy falling on the absorber surface for evaporation, the heat taken by ventilation air, and the energy stored in the solution with time. The heat loss from the surface of the regenerator is considerable and approximately 38.8% of the absorbed heat is lost around noon time. As shown in Figure 7 (bottom left and bottom right), the heat taken away by the ventilation air is very small whereas the solution stores considerable solar energy from morning till noon and releases the stored energy afternoon onwards. The concentrating performance or evaporation of water from the calcium chloride solution using the solar regenerator was also studied using the conservation of mass and energy equations for four more days, as seen in Figures 8–11.

The increase in concentration of the calcium chloride achieved was for 32.9 to 38.4%, 38.2 to 43.1%, 45.4 to 48.6% and 48.3 to 51.3% for this regenerator as a result of evaporation of 6, 4.3, 3.1, 3.0 and 2.2 kg of water, as estimated the by conservation of mass equation (Equation (5)) for day 1, day 2, day 3, day 4 and day 5, respectively. The mass of water evaporated for these days, as estimated by conservation of energy, was 5.1, 3.2, 3.1, 4.3 and 3.7 kg, respectively, for the five days. The trend in estimating the evaporation of water from the calcium chloride solution was more or less similar when estimated by the conservation of mass and energy equations as illustrated in Figures 8–11. In five days, the concentration of calcium chloride in the solution reached to 51.3%, starting from 32.9%. Thus, the open-type solar regenerator is a promising technology for liquid desiccant-based air-condition applications.

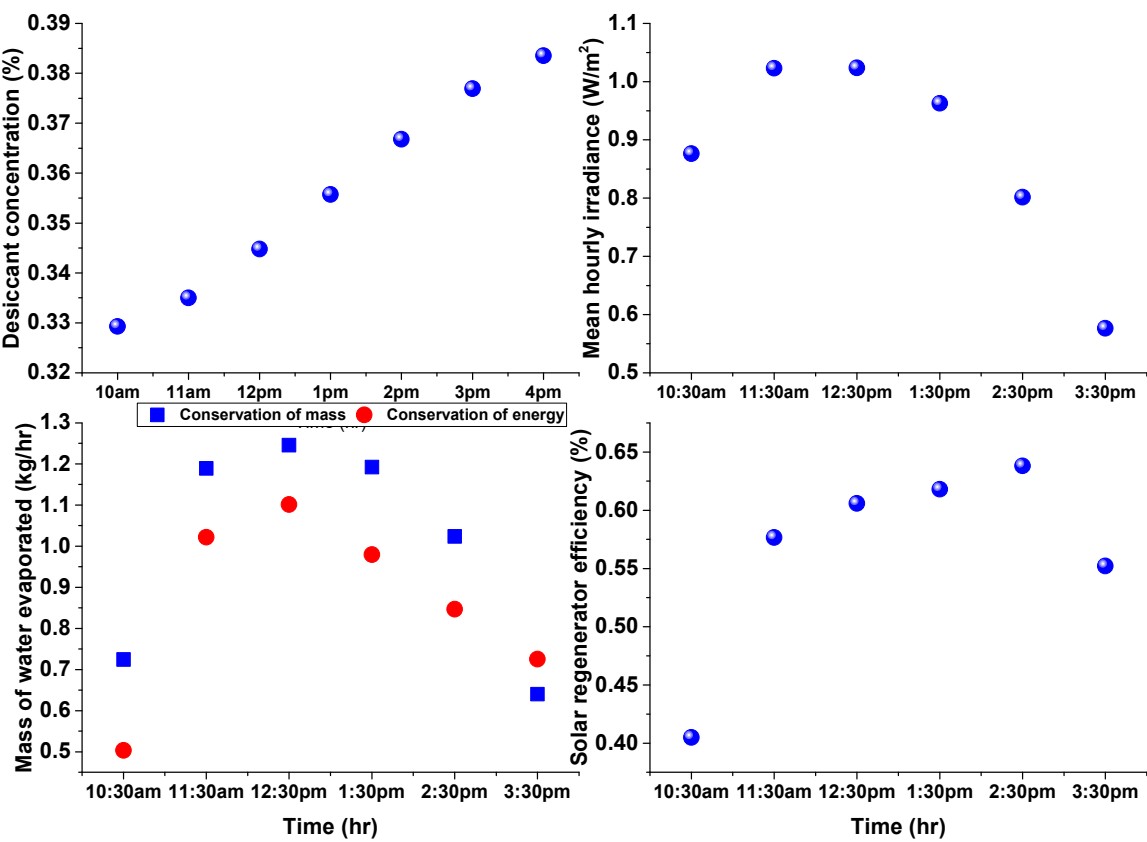

**Figure 8.** Evaporation rate of water in the solar regenerator, day 1.

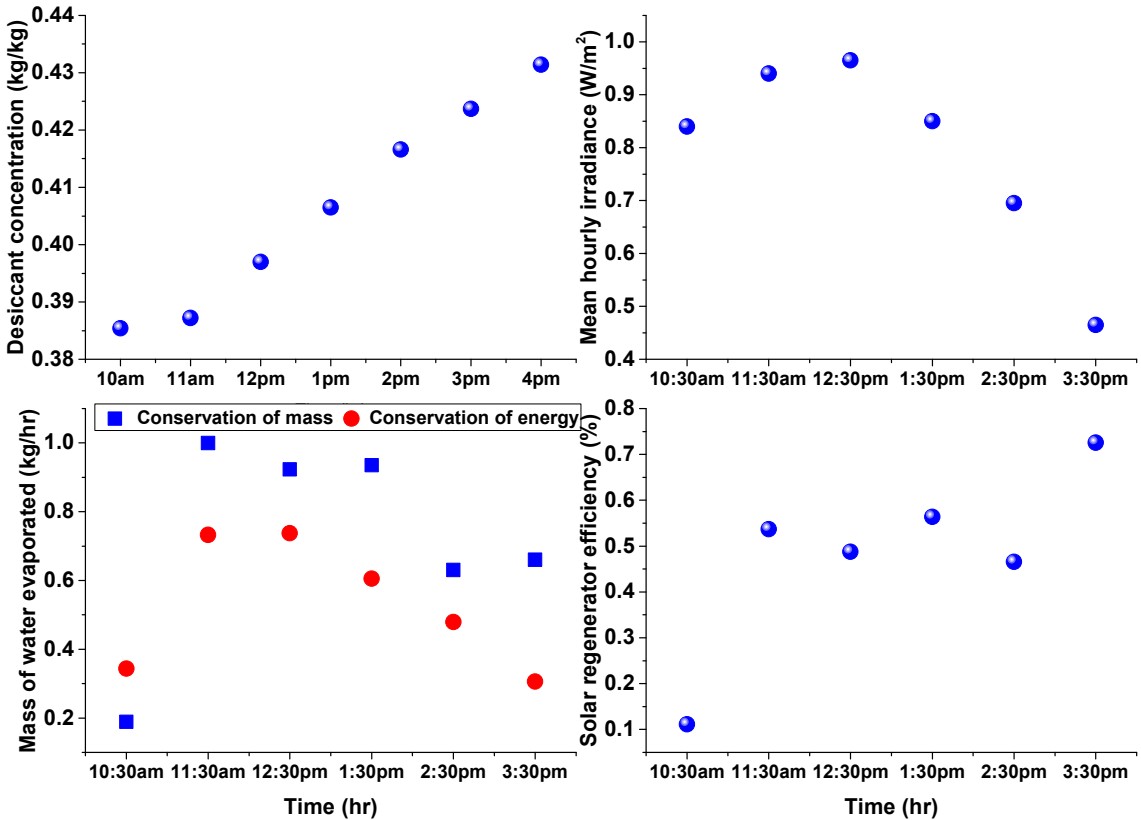

**Figure 9.** Evaporation rate of water in the solar regenerator, day 2.

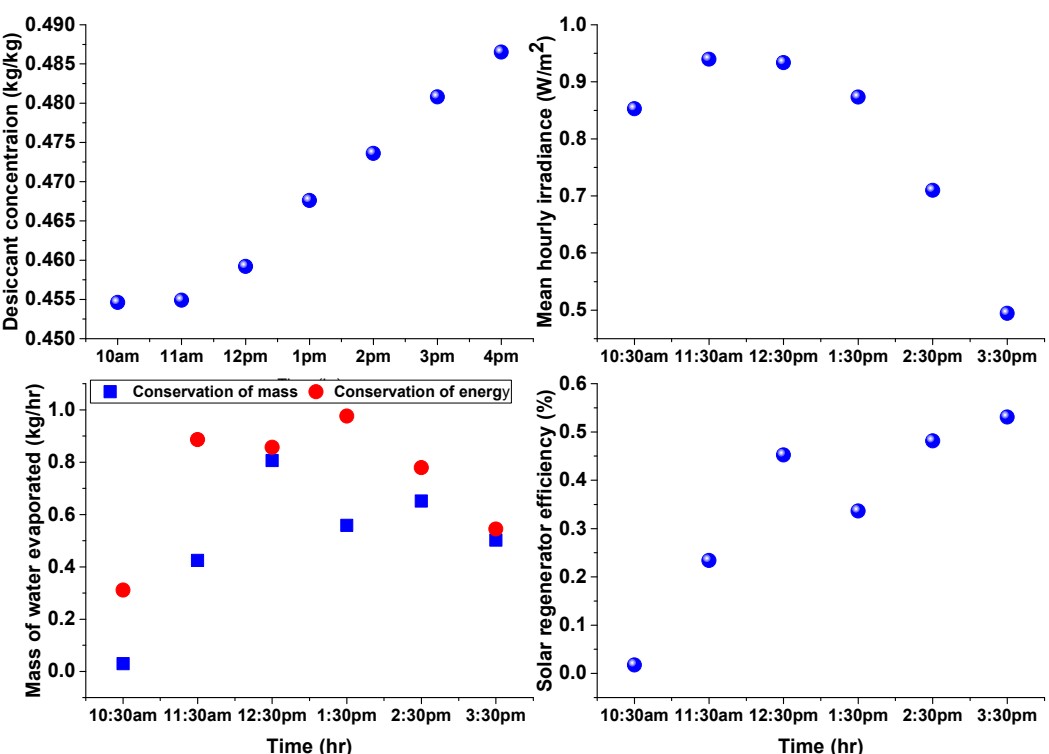

**Figure 10.** Evaporation rate of water in the solar regenerator, day 4.

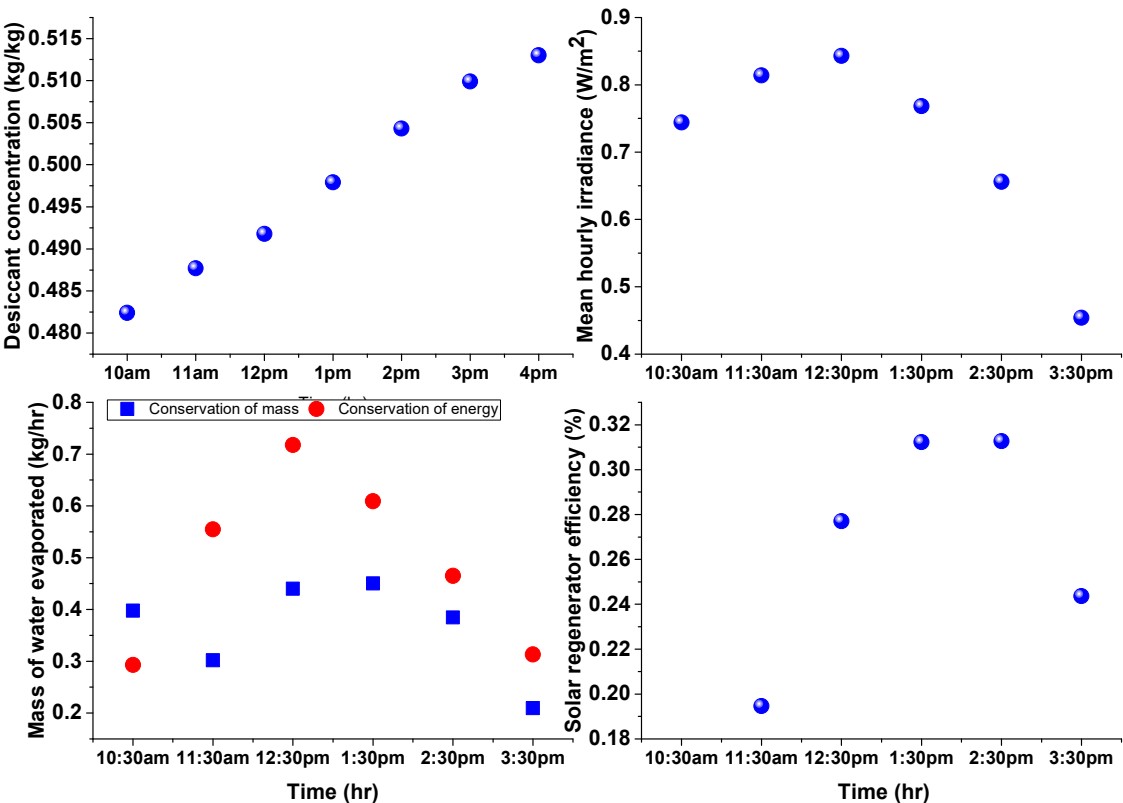

**Figure 11.** Evaporation rate of water in the solar regenerator, day 5.

As seen in Figures 8–11, there is considerable discrepancy in the hourly evaporation rate values of water as estimated by the conservation of mass and energy principles. The fluctuation in the estimation of evaporation by the conservation mass relation is due to

the inability to measure the solution density and temperature exactly after a thirty-minute time interval due to human factors since these measurements are taken manually, and the humidity ratio of water in air, which is the only resistance to mass transfer which fluctuates in the morning and afternoon time. It has been observed during experimentation that the solar regenerator operates reversely, i.e., more as air humidifier than regenerator, when the moisture transfer gradient is from the air to the liquid desiccant. This was detected by the densitometer with reduced density of the desiccant solution in the tank by the dilution of the absorbed moisture from the air. This mass transfer resistance between the moisture in the air and the liquid desiccant surface can be further studied using convective mass transfer or species transfer principles. These effects of the delayed measurement of solution density and temperature also propagate into the several relations in the energy conservation equation that depend on the solution concentration and temperature such as the specific heat, the latent heat of evaporation and the initial and final mass of the desiccant solution in the tank. There are also several other reasons that cause the persistent discrepancies that are primary related to the optical and thermal characteristics of the solar regenerator and ambient air. The optical efficiency of the solar regenerator was calculated by assuming the solar energy transmittance of glass and absorber coefficients as constants. In addition, the dust accumulation on the glass surface was also assumed to be constant. The overall thermal heat loss coefficient; in particular, the convection heat loss from the glass and between the glass and absorber were estimated by taking the constant values of the measured wind speed, which actually varies. Improvements in the data collection will improve the accuracy of the prediction of the evaporation by both conservation laws. The combined use of these conservation laws helps in studying the simulation performance and design of optimized solar regenerators when coupled with the complete liquid desiccant air conditioning system cycle since they connect the storage tank volume to the solar collection area or the length and width of the collector.

## 5. Directions for Further Research

The followings are some of the directions for furthering this research:

- Simulation of annual regeneration or evaporation performance of different size solar regenerators;
- Optimization of the size of the regenerator solar absorber area compared to the desiccant concentration and storage tank volume;
- Experimental regeneration performance study of different desiccant materials such as potassium formate and lithium chloride solution or hybrid solutions;
- Experimental performance study of the solar regenerator by directly coupling it with a liquid desiccant air conditioning system for a specified latent cooling load in the air dehumidifier;
- Experimental investigation of the thermal and optical characteristics of the solar regenerator parts such as the glass, the paints used for the absorber and the solar absorptance of different desiccant materials.

## 6. Conclusions

In this paper, the evaporation performance of a glazed free convection solar regenerator has been assessed on the basis of experimentally collected data on calcium chloride–water solution regeneration using the conservation of mass and energy principles. The regenerator is inclined and was oriented towards south. A total of twenty-five experiments were conducted by varying the quantity of the desiccant solution (mass of solution) in the supply tank, and mass flow rate of solution over the collector. The inlet concentration and temperature of the desiccant solution were varied automatically by mixing the outlet solution with the desiccant contained in the supply tank and circulating the mixture again over the surface of the collector using a centrifugal pump. The fraction of solar energy consumed to evaporate the water and the solar collection and regeneration efficiency were determined by using values of mass of water evaporated obtained by applying mass

and energy conservation principles, and the results are compared. That the evaporation estimation by the conservation of energy principle is less accurate might be due to the assumed values of the optical and thermal characteristics of the solar regenerator. Based on the performance analysis results, it can be concluded that the regeneration of the liquid desiccant using an open-type solar regenerator is feasible for coupling with liquid desiccant air cooling technologies, and the evaporation rate can be estimated with reasonable accuracy by both conservation of mass and energy principles

**Author Contributions:** Conceptualization, G.H.T. and G.M.S.A.; Software, G.H.T. and G.M.S.A.; Validation, I.A.B., S.K. and M.H.; Investigation, G.H.T.; Resources, G.H.T. and G.M.S.A.; Visualization, I.A.B., S.K. and M.H.; Writing—review and editing, I.A.B., S.K. and M.H., Writing—original draft preparation, G.H.T.; Formal Analysis, I.A.B., S.K. and M.H.; Funding Acquisition, I.A.B., S.K. and M.H.; Project Administration, I.A.B., S.K. and M.H. All authors have read and agreed to the published version of the manuscript.

**Funding:** The authors extend their appreciation to the Deanship of Scientific Research at King Khalid University for funding this work through the Small Groups Project under grant number RGP. 2/36/44.

**Institutional Review Board Statement:** Not applicable.

**Informed Consent Statement:** Not applicable.

**Data Availability Statement:** Data can be available on request.

**Acknowledgments:** The authors extend their appreciation to the Deanship of Scientific Research at King Khalid University for funding this work through the large group Research Project under grant number RGP.2/36/44.

**Conflicts of Interest:** The authors declare no conflict of interest.

**Nomenclature**

| | |
|---|---|
| $A$ | Area (m$^2$) |
| $C_P$ | Specific heat (kJ/kg K) |
| $d$ | Dust coefficient ($-$); thickness of metal strip welded along the collector left and right length (m) |
| $E_v$ | Energy of evaporation (kJ) |
| $h_{fg}$ | Convective heat transfer coefficient (kJ/kg) |
| $\Delta h_d$ | Differential enthalpy of dilution (kJ/kg) |
| $h_c$ | Convective heat transfer coefficient (W/m$^2$K) |
| $h_r$ | Radiation heat transfer coefficient (W/m$^2$K) |
| $H_{ap}$ | Total incident solar energy (kJ) |
| $H_{ab}$ | Total absorbed solar energy (kJ) |
| $I$ | Solar radiation (W/m$^2$) |
| $K$ | Thermal conductivity (W/m K) |
| $L$ | Glazing space (m); perimeter (m) |
| $Nu$ | Nusselt number (-) |
| $m$ | Mass (kg) |
| $m_d$ | Initial mass of the desiccant in the tank before starting experiment |
| $R_a$ | Rayleigh number (-) |
| $R$ | a result parameter; (varies) |
| $s$ | Shade coefficient (-) |
| $t_o$ | Starting time of experiment |
| $T$ | Temperature ($^\circ$C or K) |
| $U$ | Heat loss coefficient (W/m$^2$K); uncertainty of an instrument/a calculated variable; (varies) |
| $V$ | Wind speed (m/s); Volume (m$^3$) |

**Greek symbols**

| | |
|---|---|
| $\alpha$ | Collector absorptivity (-) |
| $\beta$ | Collector inclination (°) |
| $\xi$ | Desiccant concentration in solution (kg/kg) |
| $\rho$ | Density (kg/m$^3$) |
| $\tau$ | Glass transmissivity |
| $\sigma$ | Stefan Boltzmann constant (Wm$^{-2}$K$^{-4}$) |
| $\theta$ | Reduced temperature of water (-) |
| $\Delta t$ | Change in time (s) |
| $\eta_R$ | Collection cum regeneration efficiency (%) |
| $\nu$ | Kinematic viscosity of air (m$^2$ s$^{-2}$) |
| $\delta$ | Insulator thickness (m) |

**Subscripts**

| | |
|---|---|
| $a$ | Air |
| ab | Absorbed; absorber |
| $ai$ | Air inlet |
| ap | Aperture |
| $ao$ | Air outlet |
| $b$ | Bottom |
| $c$ | Convection |
| $d$ | Dust coefficient; |
| f | Final |
| $f_1, f_2$ | Auxiliary concentration and temperature functions |
| i | Initial; insulator |
| $g$ | Glass |
| $L$ | Loss; Length |
| $opt$ | Optical |
| $p$ | Plate |
| $r$ | Radiation |
| $R$ | Regeneration |
| $s$ | Side |
| $si$ | Initial solution; solution inlet |
| $sf$ | Final solution |
| $so$ | Solution outlet |
| $sol$ | Solution; solar |
| $sraia$ | Solar regenerator air inlet area |
| $t$ | Top; total |
| v | Vapor |

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
