# Peer review of "Estimation of Evaporation of Water from a Liquid Desiccant Solar Collector and Regenerator by Using Conservation of Mass and Energy Principles"

_sustainability, doi:10.3390/su15086520_

Round 1

Reviewer 1 Report

The presented article is a test of a liquid desiccant solar collector and regenerator. The research topic is relevant and may be of interest to specialists and researchers in the fields of solar energy and chemistry. The authors have done several experiments and obtained interesting results, however, as comments and recommendations, several points should be noted:

1. The authors should increase the literature review on the topic under consideration and consider modern research from high-ranking world publishers (most sources from the 70s-80s-90s).

2. Also, the authors should indicate in more detail the scientific novelty of the work.

3. Simultaneous citation of several sources should be avoided (for example, [6-9]) and each source should be considered separately.

4. All parameters in all formulas must be described - authors should check this.

5. The content of the work should include drawings and diagrams of the installation used, depict the principle of its operation and all its parameters.

6. What is the rationale for the installation design used - is it optimized?

7. The equipment used should be listed in a separate table and its characteristics should be described in detail.

8. Authors should add a section "Materials and Methods", where they should indicate the necessary information.

9. The parameters on pages 6 and 7 should also be placed in a separate table.

10. Graphs authors should make more colorful and revealing, which will enhance the work and attract the reader.

11. The authors should significantly expand the discussion section, where they should analyze the results in more depth.

12. Where and how do the authors plan to use the results obtained in the work?

13. The authors should add a section "Directions for further research", where they should indicate the planned work on the topic under consideration.

14. Section "Acknowledgments" is duplicated.

In general, the presented article leaves a positive impression, however, it is not without shortcomings. After eliminating these remarks and taking into account the recommendations made, the presented article may be of interest to readers of the journal "Sustainability".

Author Response

Response for Reviewer 1

Comments and Suggestions for Authors

The presented article is a test of a liquid desiccant solar collector and regenerator. The research topic is relevant and may be of interest to specialists and researchers in the fields of solar energy and chemistry. The authors have done several experiments and obtained interesting results, however, as comments and recommendations, several points should be noted:

  1. The authors should increase the literature review on the topic under consideration and consider modern research from high-ranking world publishers (most sources from the 70s-80s-90s).
  • Some more literature are added.
  1. Also, the authors should indicate in more detail the scientific novelty of the work.
  • This is addressed towards the end of the introduction.
  1. Simultaneous citation of several sources should be avoided (for example, [6-9]) and each source should be considered separately.
  • The sources addressed as [6-9] were four papers by same authors. These are now written after revision as [15,16,17,18]
  1. All parameters in all formulas must be described - authors should check this.
  • Missed descriptions are added
  1. The content of the work should include drawings and diagrams of the installation used, depict the principle of its operation and all its parameters.
  • These requests are addressed in the revision under subheading 2 with new diagrams.
  1. What is the rationale for the installation design used - is it optimized?
  • Solar liquid desiccant air conditioning system cycle has two critical components: air dehumidifier and solar regenerator. The liquid desiccant flow between these two components and performs moisture absorption in the air dehumidifier and rejection of this exact amount of moisture in the solar regenerator when the two systems operate together in a cycle. The size of the solar regenerator area should be optimized when the two systems operate in a cycle. However, in this study small solar regenerator performance was tested solely to identify the solar thermal evaporation performance over a day from context of operating the solar regenerator without the need to fulfil the exact & strict moisture rejection requirement and concentration needed at inlet to the air dehumidifier. This will provide the liberty of regenerating as much liquid desiccant (low concertation to high concentration) separately and store it for use by the air dehumidifier.
  1. The equipment used should be listed in a separate table and its characteristics should be described in detail.
  • Table 1 is added and instruments characteristics are described nearby.
  1. Authors should add a section "Materials and Methods", where they should indicate the necessary information.
  • Materials and Methods is added as subheading 2
  1. The parameters on pages 6 and 7 should also be placed in a separate table.
  • Table 2 is added
  1. Graphs authors should make more colorful and revealing, which will enhance the work and attract the reader.
  • This comment is addressed for each graphs
  1. The authors should significantly expand the discussion section, where they should analyze the results in more depth.
  • Discussion was expanded
  1. Where and how do the authors plan to use the results obtained in the work?
  • The results obtained in this work can be used for validation of existing analytical equation ( e.g. Collier, [2]) that can be used for simulating evaporation performance and optimization of the solution tank capacity verses the solar collector area (width and length) through combining the conservation of mass relation with the conservation of energy relation. Two new analytical equation (power series function and convective mass transfer) were derived and the data obtained in this work are useful for validating these relations, too. Draft papers are prepared in this regard.

  1. The authors should add a section "Directions for further research", where they should indicate the planned work on the topic under consideration.
  • Addressed as subheading 5.

  1. Section "Acknowledgments" is duplicated.
  • The duplicate is omitted

In general, the presented article leaves a positive impression, however, it is not without shortcomings. After eliminating these remarks and taking into account the recommendations made, the presented article may be of interest to readers of the journal "Sustainability".

Reviewer 2 Report

The article investigates the evaporation of water from a liquid desiccant solar collector. The research is relevant and the paper can be interesting for others. However, from my point of view improvements are neccessary before publication:

- The main contribution of the paper should be clarified at the end of the introduction.

-  Measurement uncertanties are provided for the sensors but not for the final results. In my opinion uncertainties for e.g. regenerator efficiency should be provided. 

- The deviation between the estimated mass evaporated obtained by mass fraction and energy conversation should be discussed a little more. Discussion is provided for a first case but not for the other investigations. How could the desrepance depend e.g. on operating conditions. Why even use energy conservation, if concentration can be measure more reliable and is used for efficiency calculation?

Author Response

Response for Reviewer 2

Comments and Suggestions for Authors

The article investigates the evaporation of water from a liquid desiccant solar collector. The research is relevant and the paper can be interesting for others. However, from my point of view improvements are necessary before publication:

- The main contribution of the paper should be clarified at the end of the introduction.

  • A paragraph is added.

-  Measurement uncertainties are provided for the sensors but not for the final results. In my opinion uncertainties for e.g. regenerator efficiency should be provided. 

  • The uncertainty for final peak results that is concentration, evaporated mass of water and the efficiency were reported as sample uncertainty values as “0.0014 kg/kg, 0.05 kg, and 0.02, respectively” just below the instruments uncertainties.

- The deviation between the estimated mass evaporated obtained by mass fraction and energy conversation should be discussed a little more. Discussion is provided for a first case but not for the other investigations. How could the discrepancy depend e.g. on operating conditions.

  • Further discussion in this regard is added in the results and discussion section

Why even use energy conservation, if concentration can be measure more reliable and is used for efficiency calculation?

  • The conservation of energy equation is more usefully for standardizing the construction materials optical & thermal characteristics to improve the energy efficiency since it reveals detail accounts of the collected solar energy. It is also usefully for simulation and optimization of the collector size and the solution tank for a specified cooling load for furthering the research on the complete solar liquid desiccant air conditioning system.

Round 2

Reviewer 2 Report

Manuscript was revised according to comments and can be accepted from my point of view.

Author Response

There was no new or round 2 comment from reviewer 2.
